_Report_

EMBO
Molecular Medicine

# _Plasmodium falciparum var_ genes expressed in children with severe malaria encode CIDRα1 domains

Jakob S Jespersen[1,2], Christian W Wang[1,2], Sixbert I Mkumbaye[3], Daniel TR Minja[4], Bent Petersen[5], Louise Turner[1,2], Jens EV Petersen[1,2], John PA Lusingu[1,2,4], Thor G Theander[1,2] & Thomas Lavstsen[1,2,*]

## Abstract

Most severe _Plasmodium falciparum_ infections are experienced by young children. Severe symptoms are precipitated by vascular sequestration of parasites expressing a particular subset of the polymorphic _P. falciparum_ erythrocyte membrane protein 1 (PfEMP1) adhesion molecules. Parasites binding human endothelial protein C receptor (EPCR) through the CIDRα1 domain of certain PfEMP1 were recently associated with severe malaria in children. However, it has remained unclear to which extend the EPCR-binding CIDRα1 domains epitomize PfEMP1 expressed in severe malaria. Here, we characterized the near full-length transcripts dominating the _var_ transcriptome in children with severe malaria and found that the only common feature of the encoded PfEMP1 was CIDRα1 domains. Such genes were highly and dominantly expressed in both children with severe malarial anaemia and cerebral malaria. These observations support the hypothesis that the CIDRα1-EPCR interaction is key to the pathogenesis of severe malaria and strengthen the rationale for pursuing a vaccine or adjunctive treatment aiming at inhibiting or reducing the damaging effects of this interaction.

**Keywords** CIDR; EPCR; PfEMP1; severe childhood malaria; _var_
**Subject Categories** Chromatin, Epigenetics, Genomics & Functional Genomics; Microbiology, Virology & Host Pathogen Interaction

## Introduction

The majority of children living in malaria endemic areas of Africa suffer at least one attack of severe malaria during early childhood (Goncalves _et al_, 2014). These malaria attacks caused by

_Plasmodium falciparum_ parasites are life threatening, and it is estimated that more than half a million children die each year (Murray _et al_, 2014; Snow, 2014). Most of the millions of paediatric malaria cases seeking treatment at primary health facilities are treated as outpatients, but based on a clinical assessment, the physician admits a small fraction of these children to hospital. Given adequate treatment, most of these children survive, but even at the best facilities, some die. On hospital admission, a few basic medical observations can identify children at risk of a fatal outcome. By far the majority of deaths are found in those who have respiratory distress, haemoglobin levels below 5 g/dl (severe anaemia) or are unconscious (cerebral malaria) (Marsh _et al_, 1995). These broad clinical categories do not constitute well-defined pathologies, and they often overlap in the same patient. It is clear that severe malaria is associated with several different pathological processes such as proinflammatory and procoagulant activation of endothelial cells, compromised blood–brain barrier, and vascular congestion caused by infected erythrocytes (see recent review Wassmer _et al_, 2015). At their late stages, _P. falciparum_ infected erythrocytes sequester in post-capillary venules and are not detected in peripheral blood. This is thought to be the triggering phenomenon for many of the pathological processes involved in severe malaria (MacPherson _et al_, 1985; Nguansangiam _et al_, 2007; Aguilar _et al_, 2014; Milner _et al_, 2015; Seydel _et al_, 2015; Wassmer _et al_, 2015). Immunity to the severe forms of malaria is acquired early in life, and few children suffer more than one or two life-threatening attacks (Goncalves _et al_, 2014). A fundamental question is therefore whether parasites causing severe malaria as opposed to uncomplicated disease share common traits and whether different manifestations of severe malaria can be linked to particular parasite features.

_Plasmodium falciparum_ erythrocyte membrane protein 1 (PfEMP1) are parasite-derived proteins, which anchor infected erythrocytes to receptors on the vascular lining, and by this process of sequestration the parasites avoid splenic clearance and death (Hviid & Jensen, 2015). Each _P. falciparum_ genome contains

1 Centre for Medical Parasitology, Department of Immunology & Microbiology, University of Copenhagen, Copenhagen, Denmark
2 Department of Infectious Diseases, Rigshospitalet, Copenhagen, Denmark
3 Kilimanjaro Christian Medical University College, Kilimanjaro Clinical Research Institute, Moshi, Tanzania
4 National Institute for Medical Research, Tanga Research Centre, Tanga, Tanzania
5 Centre for Biological Sequence Analysis, Technical University of Denmark, Kgs. Lyngby, Denmark
  *Corresponding author. Tel: +45 30239113; E-mail: thomasl@sund.ku.dk

~60 *var* genes that encode different PfEMP1 molecules (Baruch *et al*, 1995; Smith *et al*, 1995; Su *et al*, 1995). *Var* gene regulation ensures that only one PfEMP1 variant is expressed on the surface of each infected erythrocyte (IE) (Scherf *et al*, 1998); however, several PfEMP1 variants are expressed by the many millions IEs present in an infected individual. Accumulated evidence converge towards the hypothesis that severe malaria is caused by parasites expressing a distinct subset of PfEMP1 and that antibodies to these PfEMP1 types are the mediators of an early acquired immunity protecting children from severe malaria attacks (Bull *et al*, 1998; Nielsen *et al*, 2002; Cham *et al*, 2009, 2010; Hviid & Jensen, 2015; Turner *et al*, 2015). Under the dual evolutionary pressure to retain host receptor interaction capacity while diversifying to escape immune recognition, *var* genes have acquired extreme sequence diversity, but within a highly ordered genetic organization and composition of encoded domains that correlate with host receptor binding phenotypes (Lavstsen *et al*, 2003; Kraemer & Smith, 2006; Rask *et al*, 2010; Sander *et al*, 2014). Thus, binding to endothelial protein C receptor (EPCR) is restricted to PfEMP1 molecules belonging to a subgroup of subtelomeric group A *var* genes and a distinct group of subtelomeric group B *var* genes encoding domain cassette (DC) 8 (Turner *et al*, 2013). Other subtelomeric group B and centromeric group C *var* genes encode PfEMP1 molecules predicted to bind CD36 (Robinson *et al*, 2003). We have previously shown that parasite expression of EPCR-binding PfEMP1 is associated with severe malaria syndromes in hospitalized African children (Lavstsen *et al*, 2012; Bertin *et al*, 2013; Turner *et al*, 2013). In these studies, *var* transcript levels were quantified by real-time PCR using primers specific for gene loci unique to specific PfEMP1 domain types. This approach allowed analysis of disease association of domain types targeted by the employed primers, which due to the sequence diversity of *var* genes could not cover all domain types. Consequently, full annotation of the *var* genes expressed in children with severe malaria has not been achieved, and it has remained uncertain if domain types or domain compositions not targeted by the primers employed in the previous studies show association to severe malaria in children.

In the present paper, we have used an alternative approach (Fig 1), which allowed identification and near full-length annotation of the most prominently expressed *var* genes in individual patients. The results show that the only common characteristic of *var* transcripts in severe paediatric malaria patients was coding for domains predicted to bind EPCR.

# Results

### Annotation of var transcripts in individual patients

*Var* transcript profiles were characterized in parasites from 36 children admitted to hospital and diagnosed with severe malaria. Twenty-one patients had severe anaemia without severe cerebral impairment (Blantyre coma score > 2), nine had cerebral malaria without severe anaemia (Hb > 5 g/dl), four patients had overlapping syndromes, and two were hyperparasitemic with no other severe complication (Table 1). In addition, *var* transcripts were studied in samples from eight hospitalized children diagnosed with uncomplicated *P. falciparum* infections (Table 1). Analysis of *var* transcript distributions by DBLα-tag sequencing showed that the six most frequent *var* transcripts accounted for on average 74% of a patient's *var* transcript profile, with the 7th most frequent transcript accounting for < 1% on average. Full-length sequence information of the six most frequent transcripts in each patient was therefore pursued. Annotation of Duffy Binding-Like (DBL) and Cysteine-rich Inter Domain Region (CIDR) domains encoded by the *var* transcripts identified in the 44 individuals is shown in Appendix Fig S1.

### Comparison of var transcript based on classification of N-terminal domains

All PfEMP1s, apart from VAR2CSA and VAR3, have an N-terminal DBLα-CIDR tandem domain arrangement (Fig 1, upper panel), where the CIDR domains divide into three sequence classes, for which the human receptor binding phenotype can be predicted. CIDRα1-carrying PfEMP1s are predicted to bind EPCR (Lau *et al*, 2015), and CIDRα2-6-carrying PfEMP1s are predicted to bind CD36 (Robinson *et al*, 2003), whereas the binding phenotypes are unknown for CIDRβ, γ and δ domains. For this reason, the distribution of the top six *var* transcripts was first analysed in relation to the encoded N-terminal CIDR type. The proportion of *var* transcripts encoding CIDRα1 predicted to bind EPCR domains was higher in children with severe malaria than in children with uncomplicated disease (median proportion 54.1 vs. 7.4%, *P* = 0.005, Table 2) (Fig 2). The opposite was seen in the proportion of transcripts encoding predicted CD36-binding CIDRα2-6 domains (median 17.0 vs. 43.7%, *P* = 0.04). In children with severe malaria, EPCR-binding PfEMP1 was transcribed by both group B (DC8, carrying CIDRα1.[1/8] domains) and group A (carrying CIDRα1.[4–7] domains) EPCR-binding PfEMP1 (median transcript proportions 15.5 and 16.7%, for groups B and A, respectively). The median proportion of transcripts encoding N-terminal CIDRβ, CIDRγ or CIDRδ domains was zero in children with both severe and uncomplicated disease. Because the DBLα and N-terminal CIDR domains are found in tandem, the CIDR domain type can to some extend be predicted by the DBLα-tag sequence (see Materials and Methods). This allowed identification of putative EPCR-binding variants among the highly transcribed *var* genes where full-length sequence information could not be obtained. When these transcripts were added to the proportion of CIDRα1 encoding transcripts, the median proportion of EPCR-binding PfEMP1 transcripts in patients with severe and uncomplicated malaria was 62.3% and 7.9%, respectively (*P* = 0.002) (Table 2).

Comparison of transcript proportions in patients with severe anaemia and cerebral malaria showed that patients with severe anaemia had a higher proportion of group B type DC8 transcripts (i.e. encoding CIDRα1.[1/8] domains) than patients with cerebral malaria (median levels 16.3 and 0%, respectively *P* = 0.16) (Table 2 and Fig 2). The opposite trend was observed for group A transcripts encoding proteins predicted to bind EPCR (i.e. encoding CIDRα1.[4–7]) (11.5 and 39.6% for severe anaemia and cerebral malaria, respectively, *P* = 0.06). This difference was due to a statistically significant higher proportion of transcripts encoding CIDRα1.4 in patients with cerebral malaria (0 and 25.3% for severe anaemia and cerebral malaria, respectively, *P* = 0.02). The median ratio between group A and group B CIDRα1 transcript proportions was 1.0 in severe anaemia patients and 26.3 in patients with cerebral malaria (*P* = 0.04).

**Table 1. Clinical characteristics of patients.**

| Patients characteristics | Uncomplicated malaria[a] (*n* = 8) | Severe malaria[b] (*n* = 36) | Severe anaemia[b] (*n* = 21) | Cerebral malaria[b] (*n* = 9) |
|---|---|---|---|---|
| Age (mean ± SD, years) | 2.1 ± 1.0 | 2.3 ± 1.3 | 1.8 ± 0.9 | 3.1 ± 1.8 |
| Minimum/maximum | 0.5–3.2 | 0.15–6.2 | 0.15–3.4 | 0.6–6.2 |
| Males (%) | 37.5 | 41.7 | 47.6 | 33.3 |
| Parasitaemia (median, p/µl) | 26,355 | 65,013 | 30,275 | 61,285 |
| IQR | 10,745–49,385 | 16,870–139,615 | 12,543–113,418 | 38,693–171,483 |
| Temperature (median, °C) | 38.5 | 38.0 | 37.6 | 37.7 |
| IQR | 37.3–39.6 | 37.1–39.0 | 37.4–38.7 | 36.6–39.4 |
| Haemoglobin (mean ± SD, g/dl) | 6.9 ± 1.8 | 4.8 ± 1.8 | 3.9 ± 0.8 | 7.7 ± 1.6 |
| Minimum/maximum | 5.1–9.9 | 1.7–10.5 | 1.7–4.9 | 5.8–10.5 |
| Coma score (mean) | 3.9 | 3.6 | 4.8 | 1.3 |
| IQR | 3–5 | 2–5 | 5–5 | 0–2 |
| Minimum/maximum | 2–5 | 0–5 | 3–5 | 0–2 |
| Lactate (mean ± SD, mM) | 5.1 ± 2.3 | 6.1 ± 3.3 | 6.0 ± 3.2 | 6.6 ± 5.1 |
| Minimum/maximum | 2.4–9.0 | 1.6–17.2 | 2.0–15.2 | 1.6–17.2 |
| Glucose (mean ± SD, mM) | 4.1 ± 1.7 | 5.3 ± 2.2 | 5.0 ± 1.5 | 6.6 ± 2.8 |
| Minimum/maximum | 1.1–5.7 | 1.1–13.4 | 2.2–8.9 | 4.0–13.4 |
| Mortality (%) | 25.0 | 5.6 | 4.8 | 11.1 |

IQR, inter-quartile range.

[a]Patients admitted without fulfilling the criteria for classification as severe malaria cases. Two were septicemic and died.

[b]Severe malaria: Hb < 5 g/dl or Blantyre coma score ≤ 2 or ≥ 200,000 parasites/µl. Severe anaemia: Hb < 5 g/dl and Blantyre coma score > 2. Cerebral malaria: Hb ≥ 5 g/dl, Blantyre coma score ≤ 2 and absence of other clinical findings explaining the impaired consciousness. Four patients were diagnosed with both SA and CM, and two were hyperparasitemic with no other severe complications.

**Table 2. Proportions (%) of *var* transcripts by encoded N-terminal CIDR domain type.**

| Domain subtypes | Uncomplicated malaria (*n* = 8) Median (%) [IQR] | Severe malaria (*n* = 36) Median (%) [IQR] | *P*[a] | Severe anaemia (*n* = 21) Median (%) [IQR] | Cerebral malaria (*n* = 9) Median (%) [IQR] | *P*[a] |
|---|---|---|---|---|---|---|
| CIDRα1 | 7.4 [0–19.3] | 54.1 [19.4–72.3] | 0.005 | 63.5 [16.3–70.0] | 53.1 [37.6–67.6] | 0.60 |
| CIDRα1 or CIDRα1-prediction[b] | 7.9 [0–21.9] | 62.3 [26.5–74.7] | 0.002 | 65.7 [27.8–74.7] | 64.1 [37.6–67.6] | 0.91 |
| CIDRα1.[1/8] | 0 [0–3.6] | 15.5 [0–31.1] | 0.03 | 16.3 [0–42.2] | 0 [0–19.1] | 0.16 |
| CIDRα1.[4–7] | 3.8 [0–9.4] | 16.7 [0–47.6] | 0.08 | 11.5 [0–29.4] | 39.6 [14.0–67.6] | 0.06 |
| CIDRα1.4 | 0 [0–0] | 0 [0–20.0] | 0.05 | 0 [0–0] | 25.3 [0–39.9] | 0.02 |
| CIDRα1.[4–7]: CIDRα1.[1/8] ratio[c] | 1.0 [0.65–8.9] | 1.0 [0.34–9.1] | 0.64 | 1.0 [0.19–3.2] | 26.3 [0.69–66.6] | 0.04 |
| CIDRα2-6 | 43.7 [21.9–84.7] | 17.0 [0–33.4] | 0.04 | 14.1 [0–32.1] | 13.0 [4.0–25.0] | 0.91 |
| CIDR1β | 0 [0–0] | 0 [0–0] | 0.53 | 0 [0–0] | 0 [0–0] | 0.35 |
| CIDR1γ | 0 [0–0] | 0 [0–0] | 0.98 | 0 [0–0] | 0 [0–7.6] | 0.17 |
| CIDR1δ | 0 [0–29.8] | 0 [0–0] | 0.02 | 0 [0–0] | 0 [0–0] | 0.51 |

[a]*P*-values were calculated using Wilcoxon rank-sum test.

[b]Transcripts annotated with a CIDRα1 domain or predicted by the DBLα-tag sequence to encode a CIDRα1 domain.

[c]The patient-wise ratio between the level of transcripts encoding group A type CIDRα1 domains (CIDRα1.[4–7]) and group B type CIDRα1 domains (CIDRα1.[1/8]) was calculated as follows: ($\sum$CIDRα1.[4–7] + 0.01) : ($\sum$CIDRα1.[1/8] + 0.01).

When the *var* transcript patterns were analysed according to the DBLα domain annotation, the results resembled those from the analysis based on N-terminal CIDR annotation (Appendix Table S1). Proportions of transcripts encoding DBLα0-containing PfEMP1s predicted to bind CD36 were higher in patients with uncomplicated malaria than in those with severe malaria (43.7 vs. 20.8% for uncomplicated and severe malaria, respectively, *P* = 0.05), and the opposite trend was observed for transcripts encoding DBLα (DBLα1.[1/2/4/7] and DBLα2) linked to EPCR-binding CIDR domains (9.3 and 65.0% for uncomplicated and severe malaria,

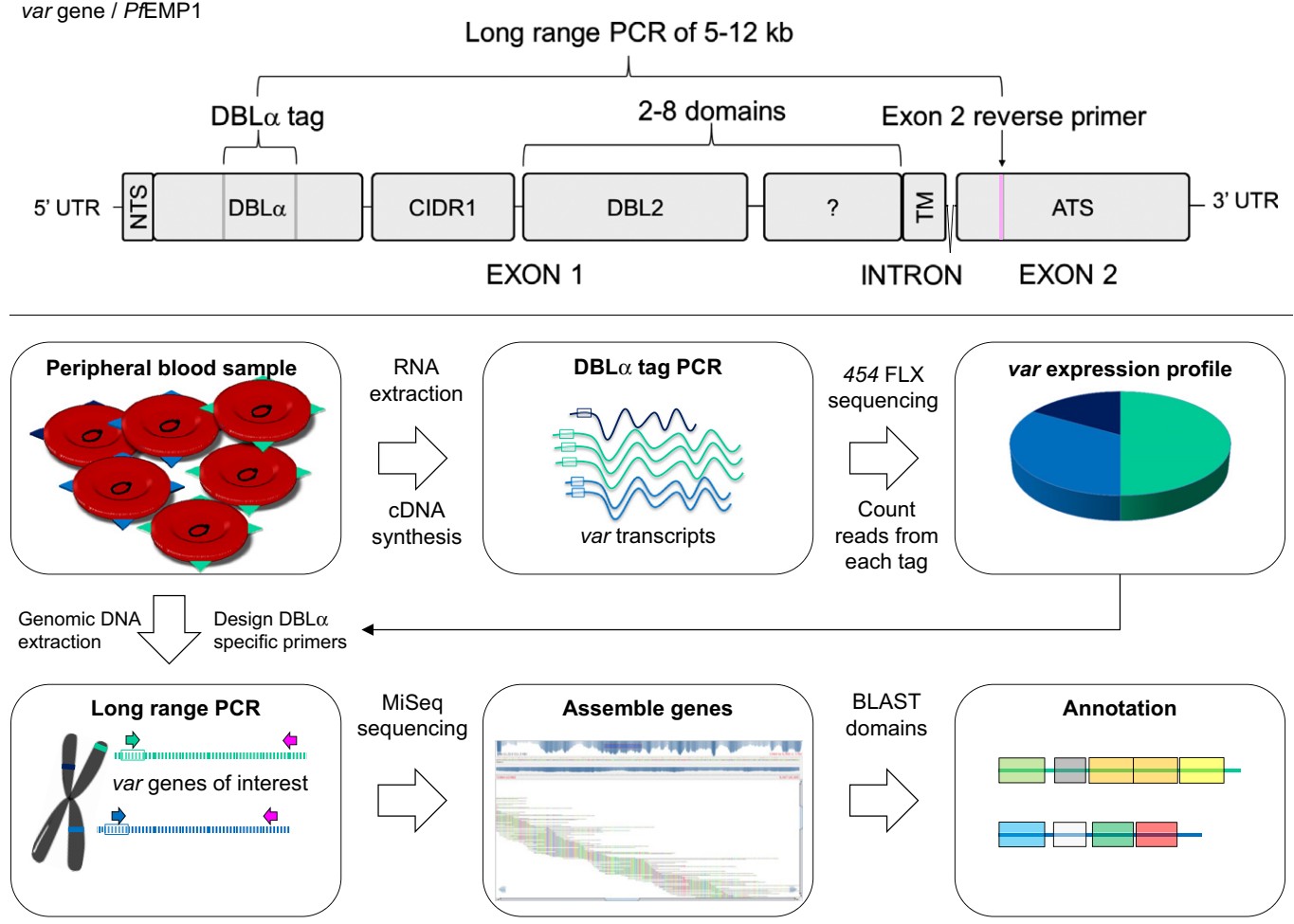

**Figure 1. *Var* gene/PfEMP1 structure and expression analysis work flow.**

*Var* gene structure (upper panel). *Var* genes consist of two exons, with exon 1 encoding the ectodomain exposed on the infected erythrocyte surface and interacting with human receptors. Exon 1 encodes the N-terminal segment (NTS), a variable number of DBL and CIDR domains and the transmembrane region (TM). Exon 2 encodes the intra-erythrocytic part of the protein, the more conserved acidic terminal segment (ATS). *Var* expression analysis work flow (lower panel). *Plasmodium falciparum* DNA and RNA was extracted from patient samples. Using cDNA from total RNA, *var* transcript profiles were generated by counting sequencing reads of PCR-amplified DBLα sequences using primers targeting semi-conserved loci flanking the DBLα-tag sequence. Based on this sequence information, single *var* gene-specific 5′ primers were designed for each of the most abundant *var* transcripts detected by the DBLα-tag PCR and paired with a 3′ primer targeting a conserved locus in exon 2 to perform a long-range PCR on genomic DNA. The resulting near full-length *var* gene fragments were sequenced, assembled and annotated.

respectively, $P = 0.005$). Proportions of transcripts encoding DBLα1.5, DBLα1.6 or DBLα1.8 domains linked to N-terminal CIDRβ, CIDRγ or CIDRδ domains were low and not associated with severe clinical status.

**Transcript patterns in individual patients**

The N-terminal CIDR types encoded in individual patients are shown in Fig 3. In most cerebral malaria patients, transcripts encoding EPCR-binding PfEMP1 were found in high proportions. In some of these patients, the profiles were dominated by group A transcripts encoding EPCR-binding PfEMP1 (e.g. patients 1918, 1914 and 1950), and in some patients, DC8 transcripts dominated (e.g. patients 1919 and 2014). In two cerebral malaria patients, the transcript profile was dominated by transcripts encoding proteins predicted to bind CD36 (patients 1996, 1995). In some patients with severe anaemia

transcripts mainly encoded DC8 PfEMP1 (e.g. patients 1965, 2120, 1920, 2142 and 2268), but in other severe anaemia patients, transcripts for EPCR-binding group A PfEMP1 dominated (e.g. patients 1939 and 2132). In a few of these patients, the dominating *var* transcripts were predicted to bind CD36 (patients 1931, 2021 and 1734). In most of the eight patients with uncomplicated malaria, the profile was dominated by transcripts encoding proteins predicted not to bind EPCR.

**Transcript patterns based on classification of other PfEMP1 domains**

In addition to the N-terminal domains, group A and DC8 PfEMP1 can contain DBLβ, DBLγ, a DBLδ-CIDR structure as well as DBLζ and DBLε domains. Some DBLβ domain types have been linked to intercellular adhesion molecule 1 (ICAM1) binding (DBLβ5 of group

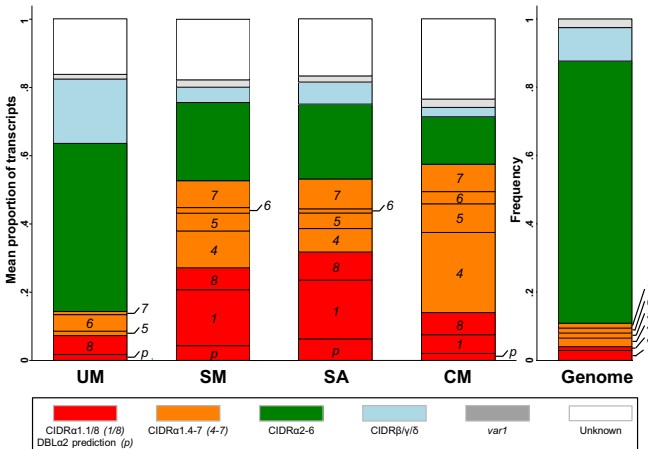

**Figure 2. Distribution of *var* transcripts and genes by encoded N-terminal CIDR domain type within patient groups and the average *Plasmodium falciparum* genome.**
Red: CIDRα1.[1/8] (DC8, binds EPCR), orange: CIDRα1.[4–7] (group A, binds EPCR), green: CIDRα2-6 (group B/C, binds CD36), blue: CIDRβ/γ/δ (group A, unknown binding phenotype), grey: *var1*, white: CIDR type not identified. Severe malaria (SM), cerebral malaria (CM), severe anaemia (SA) and uncomplicated malaria (UM).

B and DBLβ3 of group A PfEMP1 (Smith *et al*, 2000; Bengtsson *et al*, 2013). The binding specificity of the remaining C-terminal domains is uncertain, but several of these domains form domain cassettes (Rask *et al*, 2010). None of these domain types alone, in domain cassettes or domain count alone, were associated with severe malaria or severe malaria syndrome (Appendix Table S2 and Fig EV1), although the common four domain groups B and C PfEMP1 as well as PfEMP1 with DBLζ or DBLε domains were associated with uncomplicated malaria in agreement with previous observations made by quantitative PCR analysis of *var* transcript analysis (Lavstsen *et al*, 2012).

In order to investigate whether a particular subset of EPCR-binding PfEMP1 were associated with severe malaria or severe malaria syndrome, the domain composition of expressed EPCR-binding PfEMP1 was compared to the occurrence of these domains and domain compositions within EPCR-binding PfEMP1 encoded by the average parasite genome (Rask *et al*, 2010). No separate domain composition within EPCR-binding PfEMP1 was found to be associated with expression in the patients.

## Discussion

The epidemiology of severe malaria in areas of high transmission, serological typing of parasites from children with severe disease and antibody profiling in children surviving their first malaria attacks (Bull *et al*, 1998; Nielsen *et al*, 2002) suggest that severe paediatric malaria is precipitated by *P. falciparum* expressing a distinct set of PfEMP1 molecules on the surface of infected erythrocytes (Smith *et al*, 2013). This group of molecules is predicted to share features that make the parasites expressing them particularly dangerous to the host, most likely by conveying a particular host receptor binding phenotype. Different PfEMP1 molecules have been shown to bind different endothelial receptors, and mediate formation of rosettes

with uninfected erythrocytes (see review Hviid & Jensen, 2015). However, *in vitro* phenotypic profiling of parasites isolated from patients has failed to identify a binding phenotype consistently associated with severe disease outcome. It has also proved difficult to associate specific PfEMP1 sequence types to severe malaria syndromes, mainly due to large size and sequence diversity of *var* genes. Most studies have relied on characterizing expressed DBLα-tags (Kirchgatter & Portillo Hdel, 2002; Kyriacou *et al*, 2006; Normark *et al*, 2007; Falk *et al*, 2009; Warimwe *et al*, 2009) and have highlighted the importance of group A *var* genes. These studies were limited to inform only on the DBLα domain and could not distinguish between group A genes encoding CIDRα1 or other N-terminal CIDR domains, nor could they reliably discriminate between CD36-binding group B-C genes and EPCR-binding group B (DC8) genes. Other studies have employed quantitative PCR targeting a broader range of *var* gene features (Rottmann *et al*, 2006; Lavstsen *et al*, 2012; Bertin *et al*, 2013). These studies associated expression of DC8 *var* genes with children suffering from severe malaria symptoms and emphasized the particular importance of group A *var* genes encoding the CIDRα1 domain, including the DC13 variants. Later studies showed that DC8 PfEMP1 and a subset of group A PfEMP1 molecules mediate binding to EPCR through their N-terminal CIDRα1 domain (Turner *et al*, 2013; Lau *et al*, 2015), implicating EPCR-binding as an important feature of parasites causing severe malaria. However, a complete description of the domain composition of the PfEMP1 molecules expressed by parasites causing severe malaria was missing and the relative importance of different domain types including those predicted not to bind EPCR was left unanswered.

In this study, we extracted the near full-length PfEMP1 ectodomain sequence of the most prominently expressed *var* genes in 44 patients. This provides the most comprehensive characterization of *var* transcripts in patients with severe paediatric malaria to date.

The DBL and CIDR domains in PfEMP1 have an impressive but not unlimited capability to tolerate sequence mutation. DBL and CIDR domain share core structural features (Lau *et al*, 2015) and are readily divided into relatively few distinct sequence subtypes (DBLα-ζ and CIDRα-δ). Different groups of PfEMP1 molecules are characterized by having an ordered domain composition, and a tightly controlled recombination mechanism is acting to keep domain types and domain order in place (Lavstsen *et al*, 2003; Kraemer & Smith, 2006; Rask *et al*, 2010; Sander *et al*, 2014). In agreement with this, PfEMP1 receptor specificity appears to be linked to particular DBL and CIDR subtypes. The most striking example of this is the binding of PfEMP1 to EPCR by CIDRα1 domains, CD36 by CIDRα2-6 domains, and to neither (but probably yet unidentified receptors) by CIDRβ/γ/δ domains.

In each patient, several different *var* transcripts were detected, but the majority of transcripts arose from less than a handful of genes. In children with severe malaria, the most frequent transcripts encoded EPCR-binding CIDRα1 domains, these group A or group B (DC8) *var* genes dominated the *var* gene transcriptome in most patients with severe disease, and they were far more dominant than expected from the distribution of these *var* gene subtypes in the parasite genome. Conversely, transcripts encoding PfEMP1 predicted to bind CD36, which are also found in highest number in the parasite genome, dominated in the patients with uncomplicated malaria. Group A *var* transcripts encoding

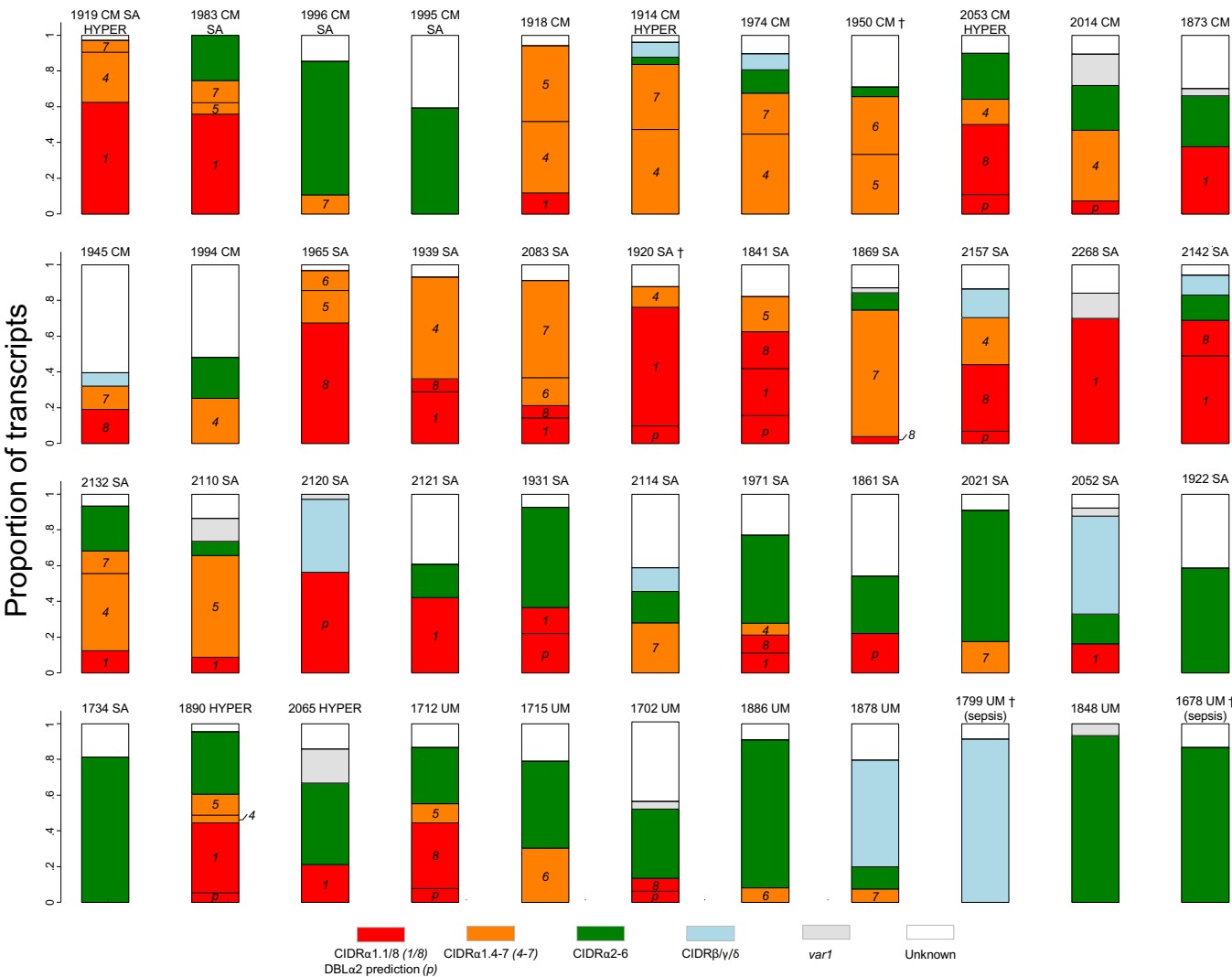

**Figure 3. Distribution of *var* transcripts by their encoded N-terminal CIDR domain type within individual patients.**
Patient numbers and diagnoses cerebral malaria (CM), severe anaemia (SA), hyperparasitemia (HYPER) and uncomplicated malaria (UM). Patients who died are indicated with (†). Red: CIDRα1.[1/8] or DBLα2 prediction (DC8, binds EPCR), orange: CIDRα1.[4–7] (group A, binds EPCR), green: CIDRα2-6 (group B/C, binds CD36), blue: CIDRβ/γ/δ (group A, unknown binding phenotype), grey: *var1*, white: CIDR type not identified.

N-terminal CIDRβ/γ/δ domains were not prominent in any patient group.

In agreement with a recent study (Abdi *et al*, 2015), there was a tendency that group A genes predicted to bind EPCR were more prominent in patients with cerebral malaria, whereas DC8 (group B) genes were more prominent in patients with severe anaemia (Table 2). However, this association was by no means absolute, as some cerebral malaria patients had dominant DC8 transcription and some severe anaemia patients were dominated by transcripts encoding group A EPCR-binding PfEMP1s. Also, all of the six subtypes of EPCR-binding CIDRα1 (1.1 and 1.4–1.8) were found highly expressed in at least some patients with severe malaria. Thus, it is likely that future vaccines aiming to protect children against severe malaria by induction of antibodies inhibiting the CIDRα1-EPCR interaction should target all EPCR-binding CIDRα1 subgroups.

In a few patients with severe malaria (~11%), < 10% of the *var* transcript profile could be ascribed transcripts identified to encode

EPCR-binding PfEMP1, indicating that severe malaria in some cases could be precipitated in the absence of EPCR-binding parasites. However, it is also possible that severe symptoms in these patients were precipitated by other pathologies than malaria. Studies from Malawi have shown that about 25% of malaria patients classified as having cerebral malaria based on clinical findings, after autopsy was concluded to have died from other causes (Taylor *et al*, 2004; Milner *et al*, 2015). The finding that some of the patients with uncomplicated malaria showed high proportions of transcripts encoding EPCR-binding PfEMP1 was expected, as these children might well have developed severe symptoms had the infection been allowed to progress without administration of antimalarial drugs.

ICAM1 binding has been described as an important binding phenotype in severe malaria (Berendt *et al*, 1989; Newbold *et al*, 1997; Ochola *et al*, 2011; Turner *et al*, 2015). ICAM1 binding has been mapped to DBLβ3 and DBLβ5 domains (Smith *et al*, 2000; Howell *et al*, 2008; Brown *et al*, 2013; Gullingsrud *et al*, 2013)

predominantly found in group A and B PfEMP1, respectively. Genes encoding potential ICAM1-binding group B PfEMP1 were expressed in some patients, but at a low level and with no association to disease severity or syndrome (Appendix Table S2 and Fig EV1). Expression of group A genes encoding DBLβ3 domains was found more frequently, and more often than expected from genomic prevalence, but with no clear association to severe disease or severe disease syndrome. However, the current study included relatively few patients and was not powered to detect traits that contribute to disease severity in a minor part of the patients with severe malaria. The molecular basis for the DBLβ-ICAM1 interaction is poorly understood, making prediction of ICAM1-binding domains uncertain. Nevertheless, our results suggest that severe malaria can be precipitated in the absence of ICAM1 binding (e.g. by EPCR-binding DC8 PfEMP1 previously found not to bind ICAM1; Avril *et al*, 2012; Claessens *et al*, 2012) and that ICAM1 binding in absence of EPCR binding (via group B-C PfEMP1 not binding EPCR; Janes *et al*, 2011) is not an important determinant of severe malaria. However, the observation that EPCR-binding group A PfEMP1 was more frequent in cerebral malaria patients compared to those with severe malaria anaemia could reflect a dual EPCR and ICAM1 binding capability of some of these PfEMP1 variants (Bengtsson *et al*, 2013). Further elucidation of the molecular determinants of PfEMP1–ICAM1 interaction and ability to predict ICAM1-binding from the amino acid sequence is required to assess the role that group A PfEMP1 binding both ICAM1 and EPCR play in the pathogenesis of cerebral malaria.

Rosetting, the binding between an infected erythrocyte and several uninfected erythrocytes, has been associated with parasites isolated from severe malaria patients (Carlson *et al*, 1990; Rowe *et al*, 1995; Heddini *et al*, 2001). This binding phenomenon may be mediated by DBLα1.[5/6/8] domains (Rowe *et al*, 1997; Chen *et al*, 1998; Angeletti *et al*, 2015) found in context with CIDRβ, CIDRγ and CIDRδ domains (Juillerat *et al*, 2011; Ghumra *et al*, 2012). Such genes were not prominently expressed in the patients with severe malaria (e.g. Fig 2). Instead, transcripts in severe malaria patients were most often encoding DBLα types found in CIDRα1 containing PfEMP1 binding EPCR. However, a clinical importance of rosetting remains unclear, as the phenotype cannot be predicted from the PfEMP1 amino acid sequence (Horata *et al*, 2009) and has also been associated with both RIFIN (Goel *et al*, 2015) and STEVOR proteins (Niang *et al*, 2014). PECAM1 binding (Treutiger *et al*, 1997) has been associated with PfEMP1 containing DC5 (Berger *et al*, 2013), but *var* transcripts encoding DC5 PfEMP1 were not found in high abundance in the studied children. Binding to IgM (Semblat *et al*, 2006, 2015; Stevenson *et al*, 2015a) and alpha-2-macroglobulin (Stevenson *et al*, 2015b) has been mapped to C-terminal PfEMP1 domains, DBLε and/or DBLζ, which exhibited higher transcript proportions in children with uncomplicated malaria (Fig EV1).

The transcript levels of *var2csa* and *var3* cannot be assessed by the DBLα-tag method used here. These two *var* genes are highly conserved and easily investigated by quantitative PCR. Neither *var2csa* nor *var3* was found highly expressed in children with paediatric malaria by quantitative PCR analysis of samples from the same patient population investigated in this study (Lavstsen *et al*, 2012).

In conclusion, the only PfEMP1 domain consistently associated with severe paediatric malaria, regardless of qualifying syndrome,

was CIDRα1 predicted to mediate EPCR binding. This raises the question whether the interaction with EPCR provokes all the various pathological processes initiated by parasite sequestration. EPCR activation of protein C plays an essential role in the regulation of coagulation, vascular inflammation and endothelial permeability (Bouwens *et al*, 2015). PfEMP1 binding to EPCR inhibits protein C conversion (Gillrie *et al*, 2015; Petersen *et al*, 2015; Sampath *et al*, 2015), and thus, parasite sequestration through EPCR engagement could directly influence pathogenic processes leading to unfavourable inflammation and coagulation events (Moxon *et al*, 2013) as well as leakage through the blood–brain barrier and brain swelling (Seydel *et al*, 2015). While the presence of sequestered parasites in the brain is believed to be key to the organ specific pathology in cerebral malaria, it is unclear whether sequestration in the bone marrow promotes development of anaemia. However, sequestration of blood stage parasites and deposition of haemozoin is a common finding in bone marrow aspirates from anaemic children living in malaria endemic areas (Aguilar *et al*, 2014). Future studies must establish if the PfEMP1 binding to EPCR in itself contributes to development of anaemia.

PfEMP1 with CIDRα1 domains is large multidomain molecules, and it is possible that other domains potentiate the parasite cytoadhesion and could contribute to pathogenesis systemically or in specific organs. It is also possible that variations of host genetic or epigenetic factors or co-infections (Hochman *et al*, 2015) exacerbate or direct disease outcome.

There is little doubt that severe malaria is a multifaceted disease governed by several host and parasite factors. However, the data presented here strongly indicate that the interaction between CIDRα1 and EPCR is the main and consistent driver for development of severe malaria.

# Materials and Methods

### Patients

As part of an effort to monitor malaria in Korogwe District, northeastern Tanzania, children under the age of five admitted at District Hospital (KDH) from August 2006 to July 2009 with the intention to treat for malaria were enrolled for this study after having received informed consent from the parents or legally acceptable representative. Experiments conformed to the principles set out in the WMA Declaration of Helsinki and the Department of Health and Human Services Belmont Report. This study was granted ethical clearance and approval by the Tanzania National Health Research Committee with reference NIMR/HQ/R.8a/Vol.IX/559. Previously, patient samples were randomly selected from a database analysed for *var* abundances using quantitative PCR (Lavstsen *et al*, 2012). Those of these samples where additional Trizol® preserved RNA were available were included in the present study. For the analysis of *var* transcripts, patients were characterized as having severe malaria if presenting with respiratory distress, Blantyre score ≤ 2, severe anaemia (< 5 g/dl) or hyperparasitemia (≥ 200,000 parasites/µl) and absence of any other disease explanatory clinical finding. Patients without any of these syndromes or with confounding clinical conditions were classified as uncomplicated malaria cases.

## Var transcript analysis strategy

First, the relative abundance of *var* transcripts in each patient was determined by cDNA DBLα-tag PCR amplification, sequencing and counting of reads (Fig 1). Two identical DBLα-tags are rarely observed in the parasite population (Barry *et al*, 2007), and the DBLα-tag PCR primers are designed to target all *var* genes, with the exception of the conserved *var2csa* and *var3* genes (Lavstsen *et al*, 2012). On average, the six most frequent DBLα-tags accounted for 74% (min–max; 38–100%) of each patient's *var* transcript profile, with an average individual proportion of 13%. To acquire comprehensive insight into the PfEMP1 types most prominently expressed in severe malaria patients, a specific 5′ primer was designed for each of the DBLα-tags of the six most abundant *var* transcripts of each patient and paired with a universal 3′ primer targeting a semi-conserved locus in exon 2 (Fig 1). These primers were then used to PCR amplify *var* gene fragments, encoding the near-complete PfEMP1 ectodomain, which in turn were sequenced and annotated. *Var* gene assemblies and transcript abundances were validated using gene-specific primers using conventional and quantitative PCR on each patient sample. A list of assembled and annotated transcripts is shown in Appendix Fig S1.

For some genes, it was not possible to amplify the full-length gene or to assemble the full sequence. For these domain un-annotated genes, the DBLα-tag sequence information was used to assign the transcript to a *var* group as follows: because the DBLα and CIDR domains always are found in tandem, the CIDR type and its binding phenotype can be predicted from the sequence of the last subdomain of DBLα, that is subdomain 3 (S3) (Lavstsen *et al*, 2012), and to a lesser extent also from DBLα subdomain 2 (S2) captured by the DBLα-tag. DBLα-tags can differentiate group A (DBLα1-tags) from group B and C (DBLα0-tags) *var* genes, but can only to some extend predict binding phenotypes of the encoded PfEMP1. While CD36 binding is exclusively found in groups B and C encoded PfEMP1, EPCR-binding PfEMP1 variants are encoded by approximately half of all group A genes as well as by a small subset (~6%) of B genes, encoding DC8. Using the added information from annotated genes in this study along with all other known full-length PfEMP1 sequences, analysis of a sequence distance tree built on the DBLα-tag sequences showed several subclusters mainly comprised of, and including most of DBLα2-tags. We used these clusters to identify likely DC8 *var* transcripts among the un-annotated *var* transcripts.

## RNA, cDNA and gDNA

Venous blood was drawn into citrate buffer at the time of patient admission. Total RNA was extracted following manufacturer's instructions from 50 to 200 μl packed RBCs lysed in 1,300 μl Trizol® RNA preserving reagent (Invitrogen) stored at −80°C. Prior to cDNA synthesis, RNA was treated with DNase I (Sigma) to eliminate any contaminating genomic DNA. The sample was considered DNA free when fluorescence was still baseline after 30 cycles of quantitative PCR (qPCR) with *seryl-tRNA synthetase* primers. cDNA was synthesized by reverse transcription using SuperScript® II and random hexamers (Invitrogen) at 25°C for 10 min, 42°C for 50 min and 70°C for 15 min. Genomic DNA was extracted from patient buffy coats by the automated Maxwell®16 system (Promega) or manually by the NucleoSpin® Blood kit (Macherey–Nagel). Although mainly patient

DNA, sufficient amounts of *P. falciparum* DNA was extracted for the PCR amplification of relevant genes.

## DBLα-tag PCR

The DBLα-tag primers (Lavstsen *et al*, 2012) were synthesized as *fusion* primers, which allows amplicons to fit directly into the Roche GS FLX Titanium emulsion PCR (Lib-L) sequencing preparation pipeline (http://www.tgac.ac.uk/uploads/Amplicon%20Fusion%20Primer%20Design%20Guidelines%20TGAC.pdf; http://454.com/downloads/my454/applications-info/APP001-2009-Lib-L-Unidirectional-Amplicons.pdf). The forward fusion primer had the following composition: 5′-sequencing adapter sequence A—barcode sequence—template-specific primer-3′, whereas the reverse primer lacked the 11 nucleotide barcode/multiplex identifier (MID) sequence and had adapter sequence B instead of A. The forward and reverse fusion primers were 64 and 53 nucleotides long, respectively. Thirteen fusion forward primers with MIDs differing by at least 3 nucleotides in the 8 first positions were used in combination with the fusion reverse primer. This strategy allowed amplicons made with different forward fusion primers to be pooled and sequenced in one batch and data to be separated according to MID.

PCR was performed with TaKaRa LA Taq polymerase (TAKARA Bio, Inc.) in 50 μl reactions mixed according to the manufacturer's recommendations (final primer concentrations: 1 μM) and the following PCR program: 96°C for 45 s followed by 30 cycles of 96°C for 10 s, 50°C for 20 s and 68°C for 20 s, and a final elongation at 72°C for 2 min.

## DBLα-tag amplicon sequencing and analysis

Amplicons were purified by gel electrophoresis and subsequent DNA extraction (E.Z.N.A.® Gel Extraction Kit, Omega-Biotek). The amplicons were sequenced unidirectionally (from adapter A) on a GS FLX system with Titanium Lib-L chemistry. Sequence reads were split according to their barcodes (allowing 1 mismatch and 0 deletions) and trimmed from the 3′ end until the minimum phred quality score was ≥ 15 (evaluated in a window of 5 and a step size of 1). Then, all reads with a median phred quality score < 20 or a read length < 250 nucleotides were discarded. These procedures were performed in a workflow designed on the Galaxy server http://usegalaxy.org/ (Goecks *et al*, 2010).

Clustering of the reads from each patient was performed by USEARCH version 5.0.144 (Edgar, 2010) (command line parameters: –uclust –id 0.9 –gapopen 1 –gapext 1 –iddef 0 –maxrejects 128 –nofastalign). An "R" script (http://www.r-project.org/) was written to parse the clustering results (the "R" packages "ShortRead" (Morgan *et al*, 2009), and "Biostrings" were used to handle the reads). In order to clean out singletons and other rare sequences prone to be PCR or sequencing artefacts, clusters containing < 1% of the total reads were discarded from the expression profile analysis. On average 14.3 ± 6.8% of the clustered reads were discarded on that account. Consensus sequences were determined from multiple alignments generated by MUSCLE (Edgar, 2004) manually edited in BioEdit version 7.0.9.0. Profiles of *var* transcripts were generated for each patient from the DBLα-tag results, with each read cluster corresponding to a unique transcript and cluster size to the relative transcript proportion originating from this gene.

## PCR amplification and amplicon sequencing of *var* exon 1

For each DBLα-tag sequence of interest, a tag-specific forward primer was designed using Primer-BLAST preferably positioned more than 100 nucleotides from the 3′ end of the tag. These were used in long-range PCRs in combination with a reverse primer universal to exon 2 described in Lavstsen *et al* (2012). Amplicons were separated by gel electrophoresis, and products of the expected size (~5–12 kb) were excised and purified (E.Z.N.A.® Gel Extraction Kit, Omega-Biotek). The DNA concentration was measured by a NanoDrop 2000 spectrophotometer (Thermo Scientific), and amplicons were pooled in batches of 10–50 in equimolar proportions. The batches were fragmentized by sonication (Bioruptor, Diagenode S.A.) and built into MID labelled sequencing libraries. Sequencing was performed with the Illumina HiSeq2000 system (100-bp single- and paired-end) or MiSeq (250-bp paired-end).

## Assembly and annotation of sequence data

The sequence data were assembled with Velvet version 1.2.03, SPAdes version 2.5.1 or RAY version 2.2.0. Data were trimmed and quality-filtered on the Galaxy server and error corrected with ALLPATHS-LG (Gnerre *et al*, 2011) prior to assembly. Assemblies were run with optimized *k*-mer sizes in the range 81–95 (Velvet) or with multiple automatically selected *k*-mer sizes (SPAdes). The target genes were identified among assembled contigs by BLAST searching the contigs against a library of all the DBLα-tag sequences targeted by long-range PCR. Every contig found to match a DBLα-tag was then checked for the annealing sequence of the tag-specific forward primer used in the long-range PCR. If the overlapping sequence of tag and contig was < 98% identical or < 100 bp and did not contain the primer sequence, the contig was considered an invalid match.

Annotation of *var* encoded PfEMP1 domains was performed with BLAST by gapped BLASTp searches (BLOSUM62 scoring matrix) in a library of 2,476 annotated *var* domains from 395 *var* genes (Rask *et al*, 2010). One DC8 transcript (Appendix Fig S1, 1702-1) had a rare polymorphism at the key EPCR-interacting residue of the CIDRα1 (validated by PCR and Sanger sequencing), which changed the conserved hydrophobic residue inserting into the hydrophobic groove of EPCR (in other sequences phenylalanine (94%), tyrosine (2%) or valine (3%)), to the negatively charged residue aspartic acid. This mutation is predicted to cause markedly reduced affinity to EPCR (Lau *et al*, 2015) and the domain was considered to have an unknown binding phenotype.

## Validation of *var* gene assemblies and determination of relative *var* transcript abundances

To validate that the assembled sequences were genuine, primer pairs were designed along the sequence and intended to PCR amplify 2- to 3-kb regions of *var* genes from genomic DNA from each specific patient. The PCR products were confirmed to have the expected size by gel electrophoresis. The primer pairs were designed to amplify regions overlapping by 50–300 bp, which also allowed using the reverse and forward primers flanking an overlapping region as qPCR primer pairs. Thus, 1–3 qPCR primer pairs were available for each assembled transcript sequence. These were used in qPCR to validate the transcript proportion reported by the DBLα-tag read count. qPCR was performed on a Rotorgene thermal cycler system (Corbett Research) with Quanti-Tect SYBR Green PCR Master Mix (Qiagen) with primer concentrations of 2 μM and one primer pair for *seryl-tRNA synthetase* as internal control as previously described (Lavstsen *et al*, 2012). Transcript levels relative to the internal control gene were calculated using the $2^{-\Delta Ct}$ method (Livak & Schmittgen, 2001).

## Data analysis and statistics

The six most abundant *var* transcripts in each patient were first identified from the distribution of unique DBLα-tag sequences. The relative distribution of transcripts between the top six genes (transcript proportions) was used for the comparison of expression of specific domains or domain combinations between patient groups. For the statistical analysis and figures presented, DBLα-tag determined transcript proportions were adjusted according to the transcript abundances reported by the qPCR analysis of annotated transcripts. Specifically, transcript proportions were calculated as $(2^{-\Delta Ct\ gene\#1})/\Sigma(2^{-\Delta Ct\ (gene\#\ 1-6)}$ X(100 − DBLα-tag determined transcript proportion (in per cent) of the un-annotated top six transcripts within the patient), and proportions of top six un-annotated transcripts were reported as determined by the DBLα-tag analysis, normalized to the total proportion of top six transcripts. The data were also analysed based on transcript abundance reported by the DBLα-tags alone. The comparison between patient groups gave almost identical results as those obtained by qPCR adjusted data. However, it should be noted that the two methods of quantification produced slightly different results within a given patient. Comparison of the qPCR adjusted and DBLα-tag profiles showed that a median of 21% transcript proportion (IQR: 13–29%) was differently assigned with each profile (range 5–67%). Also, a scatter plot comparing the results from the two approaches validated subdominant and dominant transcripts (Appendix Fig S2). The qPCR measured transcript abundances (Appendix Fig S3) were also found to be on a par with *var* transcript levels in placental malaria, involving only a single expressed *var* type, *var2csa* (Tuikue Ndam et al, 2005).

A database stratified on patient identity was constructed in Microsoft Excel and imported into Stata12, which was used for statistical analyses. Difference in transcript levels between patient groups was evaluated using Wilcoxon rank-sum tests.

## Sequence data

The sequences generated in this study have been deposited in GenBank under the following accession numbers: assembled genes merged with DBLα tag: KX154815–KX154979 and DBLα tag sequences alone: KX154980–KX155244. Sequences generated in a previous study (Lavstsen *et al*, 2012) were also included in the analysis here. These are deposited under the GenBank accession numbers JQ691636–JQ691650 (PopSet: 1024882945) and DBLα tags JQ691651–JQ691659 (PopSet: 387175217).

**Expanded View** for this article is available online.

## Acknowledgements

We are deeply grateful to the Tanzanian donors. The study was funded by the Danish International Development Agency (DANIDA), the Danish Council

## The paper explained

### Problem

More than half a million children die from malaria each year. Parasites causing life-threatening disease express a particularly virulent subset of *P. falciparum* erythrocyte membrane protein 1 (PfEMP1) molecules, mediating sequestration of infected erythrocytes in host organs. The molecular characteristic of this PfEMP1 subset is important for understanding severe malaria pathogenesis and development of a vaccine.

### Results

This study is the first to extract comprehensive sequence information on PfEMP1-encoding genes expressed in children with severe malaria. PfEMP1 with CIDRα1 domains, known to bind endothelial protein C receptor (EPCR), were consistently found expressed in both children suffering from severe malarial anaemia or cerebral malaria. No other of the many PfEMP1 domain types identified was associated with severe malaria symptoms. This points to the CIDRα1–EPCR interaction being the key driver of severe malaria pathogenesis.

### Impact

The identification of a single prominent PfEMP1 domain associated with parasites from severe malaria patients is encouraging for the prospects of elucidating the pathogenesis of malaria and developing a PfEMP1-based vaccine or adjunctive treatment aiming at inhibiting or reducing the burden of the disease.

for Independent Research, Medical Sciences (T1333-00220, 1331-00089B and the Sapere Aude program DFF–4004-00624B) as well as the Lundbeck Foundation, Augustinus Fonden, Axel Muusfeldts Fond and Grosserer L.F. Foghts Fond.

## Author contributions

JSJ, TL and TGT conceived and designed the study. JSJ, CWW, SIM, JEVP and LT conducted the experimental work. JSJ, BP and TL performed bioinformatics analyses. DTRM and JPAL acquired and analysed patient sample collections and clinical data. All authors analysed and interpreted the data, and all authors wrote the manuscript.

## Conflict of interest

The authors declare that they have no conflict of interest.

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
