## [Review Process File · EMBO Molecular Medicine]

***Plasmodium falciparum* var genes expressed in children with severe malaria encode CIDR 1 domains**

Jakob S. Jespersen, Christian W. Wang, Sixbert I. Mkumbaye, Daniel T. R. Minja, Bent Petersen, Louise Turner, Jens E. V. Petersen, John P. A. Lusingu, Thor G. Theander and Thomas Lavstsen

Corresponding author: Thomas Lavstsen, University of Copenhagen

Review timeline:

Submission date:	06 January 2016
Editorial Decision:	05 February 2016
Revision received:	15 April 2016
Editorial Decision:	05 May 2016
Revision received:	21 May 2016
Accepted:	23 May 2016

Editor: Céline Carret

Transaction Report:

1st Editorial Decision

05 February 2016

Thank you for the submission of your manuscript to EMBO Molecular Medicine. We have now heard back from the two referees who we asked to evaluate your manuscript. Although the referees find the study to be of potential interest, they also raise a number of concerns that need to be addressed in the next final version of your article.

As you will see from the comments pasted below, the referees find the data of interest. However, the manuscript should be rewritten taking care of limiting over-interpretations, providing more details, including additional references to position the study in a more accessible fashion to non-malaria experts and overall emphasising in a better way the novelty and importantly for our scope, the clinical relevance and putative insights for the translational field (see additional comments from Referee 2 when prompted for cross-commenting on the other referee report-below). Referee 2 also mentions some technical issues that must be addressed in a satisfactory way.

I look forward to seeing a revised form of your manuscript soon.

***** Reviewer's comments *****

Referee #1 (Remarks):

This manuscript examines var sequences to identify conserved domains that are associated with severe disease. Studies of natural infection and disease outcomes remain central to building disease hypothesis in severe malaria. Prior studies have shown that specific parasite sequences domains encoding the parasite ligand are associated with severe versus mild disease and that these ligands have corollary host endothelial cell receptors; They had a landmark study in PNAS demonstrating the association of Domain cassette 8 and 13 with severe malaria (EPCR binding); and other field studies have also shown an association of these EPCR binding parasites and severe malaria. This

study newly examines the complete relevant ligand sequence to determine if additional parasite sequence domains are associated with disease outcomes.

They data are presented in an easily understood way. They demonstrate that the mild malaria associated vars do not express parasite ligand sequences associated with adherence to EPCR, a receptor recently identified as important in severe malaria. They note that the domain CIDR α 1-EPCR is enriched in severe malaria parasites.

"The results of this study suggest that the presence of a CIDR α 1 predicted to bind EPCR is such a trait." Is the primary conclusion stated in paragraph one of the discussion? Isn't this already known, and what is the primary new finding from this in depth sequencing study? Would state clearly the new findings (even if they are confirming prior studies, can state that the in-depth sequence analysis eliminated other domains and confirm the importance of this region.)

"However, a precise description of the group A var genes involved in severe malaria was missing and the.." could be more specific and note that prior studies focused on specific domains, and this study now analysed the entire relevant sequence (ie precise description has no meaning")

Careful attention to the subtypes of severe malaria would be useful as the phenomena of pathologic parasite sequestration is specific to CM "A widespread sequestration of infected erythrocytes (IE) in various host organs appears to be the triggering phenomena for most of the pathological processes involved in severe malaria (5, 6)". ie this is not completely true and can examine the autopsy literature that specifies CM to be associated with sequestration.

A strong case of adherence and var transcript could be through associating var transcripts and HRP2, a plasma measurement that reflects the adherent biomass

They do examine transcript differences between CM (associated with pathologic sequestration and Severe anemia, which is not) and find some differences "showed a tendency towards a higher proportion of group B type DC8 transcripts (i.e. encoding CIDR α 1.1/1.8 domains) in patients with severe anaemia than in patients with cerebral malaria (median levels 22.9% and 0%, respectively P=0.09) (Table 2). The opposite trend was observed for group A transcripts encoding proteins predicted to bind EPCR (i.e. encoding CIDR α 1.4 - 1.7) (15.9% for severe anaemia vs. 21.1% for cerebral malaria P=0.13)." Are these differences important; is this the key to CM versus non CM severe disease, consider further discussion

But looking at the plots overall, the Severe anemia and CM look similar, would be interesting to discuss why they have similar patterns but sequestration only occurs in CM.

"The molecular basis for the DBL β -ICAM1 interaction is poorly understood" Thus they may be careful not to overstate that ICAM1 is not important binding ligand; if the binding mechanism is not known how can they state "that ICAM1 binding alone (via groupB-C PfEMP1 not binding EPCR) is not an important determinant of severe malaria." (many studies show that SM parasites adhere to ICAM-1)

Minor

Was there really 25% mortality in mild malaria (table 1)?

"Also, all of the six subtypes of CIDR α 1 (1.1 and 1.4-1.8) appeared to be involved in precipitating severe malaria." These studies are only association studies, cannot state they precipitate severe malaria; that requires functional studies would shorten the discussion, and have it relevant to this new data (can summarize the old studies quickly, state what the knowledge gap is and how this study provides new or confirmatory information)

Referee #2 (Remarks):

The manuscript addresses an important question about the functional domains of PfEMP1s expressed in severe malaria. The paper is well written and the data sound and well analysed. However there is a lack of clarity as to which "quantitative" data is presented and the supporting data for the validation of the quantitation is not presented.

It seems that the quantitative CIDR data presented in the paper was from the original endpoint per 454 sequencing. After 30 cycles of PCR it would be surprising if this quantitation were accurate. Why not use the subsequent, more robust, qPCR of the six most frequently detected genes per patient? It is not clear whether the quantitation of the other domains used the qPCR data or inferred the quantitation from the proportion of tags their cognate DBL α represented. Again the qRT-PCR data should have been used, as it would be more robust. The authors should clarify this point.

The actual comparison of the validation of the quantitative inference from the DBL α endpoint per sequencing should also be included. The median qPCR data for each gene compared to the proportion that each abundant DBL α tag represents of the total DBL α tags per patient should be presented, these could be scatter plots appended to suppl fig1. Some kind of statistical support should also be provided for the quantitative assumption.

minor comments

fig 2 is redundant with table 2 and less informative than table 2 which includes p values. I found the redundant discussion of the fig 2 results distracting as I spent time trying to figure out what was different from the previously described table 2. Table 2 could be extended to include the genomic frequency of the various CIDR types if the authors feel it is necessary, it does make it clear that expression frequency differs from the frequency in the genome.

supplementary fig 2 is the same as fig 2, it is CIDRs not DBLs

line 158 "transcripts encoding DBL α s linked to EPCR-binding CIDR domains" these should be named in the text as per the labelling in the table.

line 159 I couldn't see DBL α 1.7 in the table

line 177, the information about the non-severe malaria mortality should be provided as a footnote with table 1

suppl fig 3A should include error bars for std deviation, asterisks following x axis labels are not defined

Additional comment from Referee 2 as a response to cross-commenting request on referee 1 review:

"The novelty lies in the identification of the coding sequence for the entire extracellular portion of the most abundantly expressed PfEMP1s. Thus the study could have identified any additional PfEMP1 domains associated with severe disease. However the study did not find additional domains so its importance for clinical medicine lies in proving that CIDR α 1 domains alone are associated with severe disease. If correct this could have significant impact on anti-disease malaria vaccine design. However additional details and statistical support are required to prove that the quantitative analysis was robust.

Previous studies used sets of primers to various domains designed from published genomes and therefore could reasonably be suspected of missing novel PfEMP1 sequences that were absent from the reference genomes. Precedent exists for this concern as the initial candidates for adhesion to CSA in pregnancy malaria were incorrectly identified using just such an approach. Thus this study extends previous findings implicating CIDR α 1 in severe disease.

I agree that linking biomass via HRP to PfEMP1 domains is a good idea for strengthening associations with sequestration. I also agree that the assumption that severe malaria anemia is associated with sequestration is questionable. I disagree that only post-mortem associations between sequestration and cerebral malaria have been reported. Nguansangiam et al Trop Med Int Health 2007 reported associations between sequestration in the kidneys and fatal, non cerebral malaria with acute renal failure and MacPherson et al Am J Pathol 1985 reported sequestration in multiple tissues in non-cerebral fatal malaria, albeit sequestration was greatest in the brain in both cerebral and non-cerebral malaria. However I do agree that a more thorough revision of the literature surrounding sequestration in non cerebral, severe malaria would be useful."

Referee #1 (Remarks):

This manuscript examines var sequences to identify conserved domains that are associated with severe disease. Studies of natural infection and disease outcomes remain central to building disease hypothesis in severe malaria. Prior studies have shown that specific parasite sequences domains encoding the parasite ligand are associated with severe versus mild disease and that these ligands have corollary host endothelial cell receptors; They had a landmark study in PNAS demonstrating the association of Domain cassette 8 and 13 with severe malaria (EPCR binding); and other field studies have also shown an association of these EPCR binding parasites and severe malaria. This study newly examines the complete relevant ligand sequence to determine if additional parasite sequence domains are associated with disease outcomes.

The data is presented in an easily understood way. They demonstrate that the mild malaria associated vars do not express parasite ligand sequences associated with adherence to EPCR, a receptor recently identified as important in severe malaria. They note that the domain CIDR α 1 (EPCR binding) is enriched in severe malaria parasites.

"The results of this study suggest that the presence of a CIDR α 1 predicted to bind EPCR is such a trait." Is the primary conclusion stated in paragraph one of the discussion? Isn't this already known, and what is the primary new finding from this in depth sequencing study? Would state clearly the new findings (even if they are confirming prior studies, can state that the in-depth sequence analysis eliminated other domains and confirm the importance of this region.)

Author Response: We agree with the reviewer. The sentence has been removed, and the main conclusion is only outlined in the concluding remark of the discussion. The abstract has also been changed to clarify novelty.

"However, a precise description of the group A var genes involved in severe malaria was missing and the..." could be more specific and note that prior studies focused on specific domains, and this study now analysed the entire relevant sequence (ie precise description has no meaning")

Author Response: The sentence has been changed as suggested:

"However, a complete description of the domain composition of the PfEMP1 molecules expressed by parasites causing severe malaria was missing and the relative importance of different domain types including those predicted not to bind EPCR was left unanswered."

Careful attention to the subtypes of severe malaria would be useful as the phenomena of pathologic parasite sequestration is specific to CM "A widespread sequestration of infected erythrocytes (IE) in various host organs appears to be the triggering phenomena for most of the pathological processes involved in severe malaria (5, 6)" . ie this is not completely true and can examine the autopsy literature that specifies CM to be associated with sequestration.

A strong case of adherence and var transcript could be through associating var transcripts and HRP2, a plasma measurement that reflects the adherent biomass.

Author Response: It is characteristic for all P. falciparum malaria infections that peripheral blood smears only contain ring stage parasites. Since essentially only ring stage parasites are detected in peripheral blood, the parasites detected in the patients arose from parasites that were sequestered a few hours prior to blood sampling. With a median parasite density of 30,000 parasites per microliter, a child blood volume of 0.5-1 l and an effective multiplication rate of 15), this corresponds to that the parasites in the peripheral blood of an average SA child arose from >1 billion sequestered parasites. The equivalent number for the CM patients would be twice that. Sequestration of a huge parasite biomass is not unique to cerebral malaria as also evident from recent epidemiological studies (Hendriksen (Goncalves et al., 2014; Hendriksen et al., 2012), and as highlighted by reviewer 2 tissue sequestration is also documented by "Nguansangiam et al Trop Med Int Health 2007" which reported association between sequestration in the kidneys and fatal, non-cerebral malaria with acute renal failure and by "MacPherson et al Am J Pathol 1985" which

reported sequestration in multiple tissues in non-cerebral fatal malaria, albeit sequestration was greatest in the brain in both cerebral and non-cerebral malaria". More recent autopsy studies focused on cerebral malaria (Milner, Jr. et al., 2015; Milner, Jr. et al., 2014), but reported parasite sequestration in four patients with severe malarial anemia (SA).

Retinopathy, which recently have been shown to correlate well with parasite sequestration in the brain (Barrera et al., 2015), is also found in SA patients albeit more severe in CM patients (Beare et al., 2004; Essuman et al., 2010). Finally, a recent study in which bone marrow was aspirates from 290 anemic children in Mozambique showed that late blood stage parasite sequestration in the bone marrow is common and that haemozoin deposition in bone marrow is associated with severity of anaemia (Aguilar et al., 2014)

The total parasite biomass in a patient consists of the circulating stages and the hidden sequestered parasites, and therefore biomass determined by detection of peripheral parasites may be biased by the sampling time. Comparison of HRP2 plasma levels and the level of var gene transcripts in individual patients could potentially be interesting, but we believe that the reported parasite densities gives a reasonable comparison of the total parasite biomass when comparing clinically defined groups of patients.

The pathogenesis of SA is complex and involves clearance of both infected and un-infected erythrocytes and a dysfunctional erythropoiesis^(Perkins et al., 2011). We agree that evidence that sequestration per se drive pathogenesis is lacking, and our data cannot answers this.

They do examine transcript differences between CM (associated with pathologic sequestration and Severe anemia, which is not) and find some differences "showed a tendency towards a higher proportion of group B type DC8 transcripts (i.e. encoding CIDR α 1.1/1.8 domains) in patients with severe anaemia than in patients with cerebral malaria (median levels 22.9% and 0%, respectively P=0.09) (Table 2). The opposite trend was observed for group A transcripts encoding proteins predicted to bind EPCR (i.e. encoding CIDR α 1.4 - 1.7) (15.9% for severe anaemia vs. 21.1% for cerebral malaria P=0.13)." Are these differences important; is this the key to CM versus non CM severe disease, consider further discussion

But looking at the plots overall, the Severe anemia and CM look similar, would be interesting to discuss why they have similar patterns but sequestration only occurs in CM.

Author Response: We agree that the profiles are overlapping, but as argued above there is no doubt that sequestration is a prominent feature of other forms of severe malaria (only rings are detected in peripheral blood). Nevertheless it is interesting and perhaps surprising that parasites causing very different symptomatology not differ more markedly in phenotype. However earlier studies of var gene transcripts in patients are in line with our findings (Abdi et al., 2015; Amulic et al., 2009; Bertin et al., 2013; Kirchgatter and Portillo, 2002; Kyriacou et al., 2006; Lavstsen et al., 2012; Normark et al., 2007; Warimwe et al., 2009). *Epidemiological studies show that individuals in endemic areas acquire immunity to all types of severe malaria after having survived 1-3 severe malaria. This indicates that immunity to severe malaria is not syndrome-specific and that the target of this antibody mediated immunity is conserved* (Goncalves et al., 2014; Hviid, 2010)

In order to correct our statements on sequestration in and development of SA and to expand the discussion on the similar var expression patterns in CM and SA, the comment on parasite sequestration in the introduction has been changed to:

"At their late stages, *P. falciparum* infected erythrocytes sequester in post capillary venules and are not detected in peripheral blood. This is thought to be the triggering phenomena for many of the pathological processes involved in severe malaria (5-10)"

And in the concluding remark of the discussion:

"In conclusion, the only PfEMP1 domain consistently associated with severe paediatric malaria, regardless of qualifying syndrome, was CIDR α 1 predicted to mediate EPCR binding. This raises the question whether the interaction with EPCR provoke all the various pathological processes initiated by parasite sequestration. EPCR activation of protein C plays an essential role in the regulation of coagulation, vascular inflammation, and endothelial permeability (67). PfEMP1 binding to EPCR

inhibits protein C conversion (68-70) and thus parasite sequestration through EPCR engagement could directly influence pathogenic processes leading to unfavourable inflammation and coagulation events (71) as well as leakage through the blood-brain barrier and brain swelling (10). While the presence of sequestered parasites in the brain is believed to be key to the organ specific pathology in cerebral malaria, it is unclear whether sequestration in the bone marrow promotes development of anaemia. However, sequestration of blood stage parasites and deposition of haemozoin is a common finding in bone marrow aspirates from anaemic children living in malaria endemic areas (7). Future studies must establish if the PfEMP1 binding to EPCR in itself contributes to development of anaemia “

Following the re-analysis of the var transcript proportions using the qPCR data (as suggested by reviewer 2), the difference in expression levels of DC8 vs. group A EPCR binders between CM and SA patients has become clearer. We agree that this observation is interesting and warrants more attention than initially given. Thus, the statement below has been added to the Results section, and the implications on understanding CM vs SA is dealt with in the Discussion as given in the next Author response:

“Comparison of transcript proportions in patients with severe anaemia and cerebral malaria showed that patients with severe anaemia had a higher proportion of group B type DC8 transcripts (*i.e.* encoding CIDR α 1.1/1.8 domains) than patients with cerebral malaria patients (median levels 16.3% and 0%, respectively P=0.16) (Table 2 and Figure 2). The opposite trend was observed for group A transcripts encoding proteins predicted to bind EPCR (*i.e.* encoding CIDR α 1.4 - 1.7) (11.5% and 39.6% for severe anaemia and cerebral malaria, respectively, P=0.06). This difference was due to a statistically significant higher proportion of transcripts encoding CIDR α 1.4 in patients with cerebral malaria (0% and 25.3% for severe anaemia and cerebral malaria, respectively, P=0.02). The median ratio between group A and group B CIDR α 1 transcript proportions were 1.0 in severe anaemia patients and 26.3 in patients with cerebral malaria (P=0.04).“

"The molecular basis for the DBL β -ICAM1 interaction is poorly understood" Thus they may be careful not to overstate that ICAM1 is not important binding ligand; if the binding mechanism is not known how can they state "that ICAM1 binding alone (via group B-C PfEMP1 not binding EPCR) is not an important determinant of severe malaria." (many studies show that SM parasites adhere to ICAM-1)

Author Response: We have rewritten this section in the discussion to reflect the referee concern and the more clear association between group A transcripts and CM which appeared after applying the more precise qPCR based quantification method.

The paragraph now reads:

“The molecular basis for the DBL β -ICAM1 interaction is poorly understood, making prediction of ICAM1-binding domains uncertain. Nevertheless, our results suggest that severe malaria is precipitated in the absence of ICAM1 binding (e.g. by EPCR-binding DC8 PfEMP1 previously found not to bind ICAM1 (47, 48)) and that ICAM1 binding in absence of EPCR binding (via group B-C PfEMP1 not binding EPCR (49)) is not an important determinant of severe malaria. However, the observation that EPCR binding group A PfEMP1 was more frequent in cerebral malaria patients compared to those with severe malaria anaemia, could reflect a dual EPCR and ICAM1 binding capability of some of these PfEMP1 variants (31). Further elucidation of the molecular determinants of PfEMP1-ICAM1 interaction and ability to predict ICAM1-binding from the amino-acid sequence is required to assess the role that group A PfEMP1 binding both ICAM1 and EPCR play in the pathogenesis of cerebral malaria”

Minor

Was there really 25% mortality in mild malaria (table 1)?

Author Response: Yes two patients severely ill from septicemia was included as uncomplicated malaria, because they were carrying P. falciparum parasites. Both of these patients died. This is mentioned in the footnote to table 1.

"Also, all of the six subtypes of CIDR α 1 (1.1 and 1.4-1.8) appeared to be involved in precipitating severe malaria." These studies are only association studies, cannot state they precipitate severe malaria; that requires functional studies

Author Response: The sentence has been changed to:

“Also, all of the six subtypes of EPCR-binding CIDR α 1 (1.1 and 1.4-1.8) were found highly expressed in at least some patients with severe malaria”

I would shorten the discussion, and have it relevant to this new data (can summarize the old studies quickly, state what the knowledge gap is and how this study provides new or confirmatory information)

Author Response: The first paragraph of the Discussion has been condensed throughout. Please see MS text.

Referee #2 (Remarks):

The manuscript addresses an important question about the functional domains of PfEMP1s expressed in severe malaria. The paper is well written and the data sound and well analysed. However there is a lack of clarity as to which "quantitative" data is presented and the supporting data for the validation of the quantitation is not presented.

It seems that the quantitative CIDR data presented in the paper was from the original endpoint PCR 454 sequencing. After 30 cycles of PCR it would be surprising if this quantitation were accurate. Why not use the subsequent, more robust, qPCR of the six most frequently detected genes per patient? It is not clear whether the quantitation of the other domains used the qPCR data or inferred the quantitation from the proportion of tags their cognate DBL α represented. Again the q-RT-PCR data should have been used, as it would be more robust. The authors should clarify this point.

The actual comparison of the validation of the quantitative inference from the dbla endpoint pcr sequencing should also be included. The median q pcr data for each gene compared to the proportion that each abundant dbl alpha tag represents of the total dbla tags per patient should be presented, these could be scatter plots appended to suppl fig1. Some kind of statistical support should also be provided for the quantitative assumption.

Author Response: We agree with the reviewer that the qPCR measurements are to be considered a more accurate quantification than the DBL α tag PCR sequencing analysis. For this reason we have re-analysed the data and redrawn the figures using relative transcript proportions determined by qPCR.

All main observations and conclusions are similar, albeit now with a stronger association between EPCR binding PfEMP1 and severe malaria, and with a clearer differential transcript proportion of EPCR binding group A between cerebral malaria and severe anaemia patient groups. (Table 2, and Figure 2). Please see answers to rev. 1 for added comments in results and discussion sections.

Minor comments

Fig 2 is redundant with table 2 and less informative than table 2 which includes p values. I found the redundant discussion of the fig 2 results distracting as I spent time trying to figure out what was different from the previously described table 2. Table 2 could be extended to include the genomic frequency of the various CIDR types if the authors feel it is necessary, it does make it clear that expression frequency differs from the frequency in the genome.

supplementary fig 2 is the same as fig 2, it is cidrs not dbls

Author Response: The data in Figure 2 and Table 2 overlap. However, we feel that the figure conveys the data in an easy comprehensible way. We have exploited this to now show in Figure 2 only, the more detailed information on differences in CIDR α 1 subtype expression between patient groups. This also aids the discussion on the differences between CM and SA, which became more pronounced and statistically significant with the re-analysis. Please see answers to rev. 1 for added comments in results and discussion sections.

Supplementary figure 2, it did show DBL α data, but the legend was wrong. However, the data in this figure is also available in suppl. Table 1, and the figure has been removed.

line 158 "transcripts encoding DBL α linked to EPCR-binding CIDR domains" these should be named in the text as per the labelling in the table. line 159 I couldn't see DBL α 1.7 in the table

Author Response: Text has been corrected accordingly. The DBL α 1.7 was meant to be DBL α 1.8.

line 177, the information about the non-severe malaria mortality should be provided as a footnote with table 1

Author Response: This has been done.

suppl fig 3A should include error bars for std deviation, asterisks following x axis labels are not defined

Author Response: As the data are not normally distributed, the median and 25/75% percentiles are given.

Additional comment from Referee 2, cross-commenting on referee 1 review:

The novelty lies in the identification of the coding sequence for the entire extracellular portion of the most abundantly expressed PfEMP1s. Thus the study could have identified any additional PfEMP1 domains associated with severe disease. However the study did not find additional domains so its importance for clinical medicine lies in proving that CIDR α 1 domains alone are associated with severe disease. If correct this could have significant impact on anti-disease malaria vaccine design. However additional details and statistical support are required to prove that the quantitative analysis was robust.

Previous studies used sets of primers to various domains designed from published genomes and therefore could reasonably be suspected of missing novel PfEMP1 sequences that were absent from the reference genomes. Precedent exists for this concern as the initial candidates for adhesion to CSA in pregnancy malaria were incorrectly identified using just such an approach. Thus this study extends previous findings implicating CIDR α 1 in severe disease.

I agree that linking biomass via HRP to PfEMP1 domains is a good idea for strengthening associations with sequestration. I also agree that the assumption that severe malaria anemia is associated with sequestration is questionable. I disagree that only post-mortem associations between sequestration and cerebral malaria have been reported. Nguansangiam et al Trop Med Int Health 2007 reported associations between sequestration in the kidneys and fatal, non cerebral malaria with acute renal failure and MacPherson et al Am J Pathol 1985 reported sequestration in multiple tissues in non-cerebral fatal malaria, albeit sequestration was greatest in the brain in both cerebral and non-cerebral malaria. However I do agree that a more thorough revision of the literature surrounding sequestration in non cerebral, severe malaria would be useful.

Author Response: Please see response to the topic under reviewer 1.

Author Response: We would like to thank both reviewers for constructive suggestions and relevant critique.

REFERENCES

1. Abdi AI, Kariuki SM, Muthui MK, Kivisi CA, Fegan G, Gitau E, Newton CR, and Bull PC (2015) Differential *Plasmodium falciparum* surface antigen expression among children with Malarial Retinopathy. *Sci Rep*, **5**, 18034.
2. Aguilar R, Moraleda C, Achtman AH, Mayor A, Quinto L, Cistero P, Nhabomba A, Macete E, Schofield L, Alonso PL, and Menendez C (2014) Severity of anaemia is associated with bone marrow haemozoin in children exposed to *Plasmodium falciparum*. *Br J Haematol*, **164**, 877-887.

3. Amulic B, Salanti A, Lavstsen T, Nielsen MA, and Deitsch KW (2009) An upstream open reading frame controls translation of var2csa, a gene implicated in placental malaria. *PLoS Pathog*, **5**, e1000256.
4. Barrera V, Hiscott PS, Craig AG, White VA, Milner DA, Beare NA, MacCormick IJ, Kamiza S, Taylor TE, Molyneux ME, and Harding SP (2015) Severity of retinopathy parallels the degree of parasite sequestration in the eyes and brains of malawian children with fatal cerebral malaria. *J Infect Dis*, **211**, 1977-1986.
5. Beare NA, Southern C, Chalira C, Taylor TE, Molyneux ME, and Harding SP (2004) Prognostic significance and course of retinopathy in children with severe malaria. *Arch Ophthalmol*, **122**, 1141-1147.
6. Bertin GI, Lavstsen T, Guillonneau F, Doritchamou J, Wang CW, Jespersen JS, Ezimegnon S, Fievet N, Alao MJ, Lalya F, Massougboji A, Ndam NT, Theander TG, and Deloron P (2013) Expression of the domain cassette 8 Plasmodium falciparum erythrocyte membrane protein 1 is associated with cerebral malaria in Benin. *PLoS One*, **8**, e68368.
7. Essuman VA, Ntim-Amponsah CT, Astrup BS, Adjei GO, Kurtzhals JA, Ndanu TA, and Goka B (2010) Retinopathy in severe malaria in Ghanaian children--overlap between fundus changes in cerebral and non-cerebral malaria. *Malar J*, **9**, 232.
8. Goncalves BP, Huang CY, Morrison R, Holte S, Kabyemela E, Prevots DR, Fried M, and Duffy PE (2014) Parasite burden and severity of malaria in Tanzanian children. *N Engl J Med*, **370**, 1799-1808.
9. Hendriksen IC, Mwanga-Amumpaire J, von SL, Mtove G, White LJ, Olaosebikan R, Lee SJ, Tshefu AK, Woodrow C, Amos B, Karema C, Saiwaew S, Maitland K, Gomes E, Pan-Ngum W, Gesase S, Silamut K, Reyburn H, Joseph S, Chotivanich K, Fanello CI, Day NP, White NJ, and Dondorp AM (2012) Diagnosing severe falciparum malaria in parasitaemic African children: a prospective evaluation of plasma PfHRP2 measurement. *PLoS Med*, **9**, e1001297.
10. Hviid L (2010) The role of Plasmodium falciparum variant surface antigens in protective immunity and vaccine development. *Hum Vaccin*, **6**, 84-89.
11. Kirchgatter K and Portillo HA (2002) Association of severe noncerebral Plasmodium falciparum malaria in Brazil with expressed PfEMP1 DBL1 alpha sequences lacking cysteine residues. *Mol Med*, **8**, 16-23.
12. Kyriacou HM, Stone GN, Challis RJ, Raza A, Lyke KE, Thera MA, Kone AK, Doumbo OK, Plowe CV, and Rowe JA (2006) Differential var gene transcription in Plasmodium falciparum isolates from patients with cerebral malaria compared to hyperparasitaemia. *Mol Biochem Parasitol*, **150**, 211-218.
13. Lavstsen T, Turner L, Saguti F, Magistrado P, Rask TS, Jespersen JS, Wang CW, Berger SS, Baraka V, Marquard AM, Seguin-Orlando A, Willerslev E, Gilbert MT, Lusingu J, and Theander TG (2012) Plasmodium falciparum erythrocyte membrane protein 1 domain cassettes 8 and 13 are associated with severe malaria in children. *Proc Natl Acad Sci U S A*, **109**, E1791-E1800.
14. Milner DA, Jr., Lee JJ, Frantzreb C, Whitten RO, Kamiza S, Carr RA, Pradham A, Factor RE, Playforth K, Liomba G, Dzamalala C, Seydel KB, Molyneux ME, and Taylor TE (2015) Quantitative Assessment of Multiorgan Sequestration of Parasites in Fatal Pediatric Cerebral Malaria. *J Infect Dis*, **212**, 1317-1321.
15. Milner DA, Jr., Whitten RO, Kamiza S, Carr R, Liomba G, Dzamalala C, Seydel KB, Molyneux ME, and Taylor TE (2014) The systemic pathology of cerebral malaria in African children. *Front Cell Infect Microbiol*, **4**, 104.

16. Normark J, Nilsson D, Ribacke U, Winter G, Moll K, Wheelock CE, Bayarugaba J, Kironde F, Egwang TG, Chen Q, Andersson B, and Wahlgren M (2007) PfEMP1-DBL1alpha amino acid motifs in severe disease states of *Plasmodium falciparum* malaria. *Proc Natl Acad Sci U S A*, **104**, 15835-15840.
17. Perkins DJ, Were T, Davenport GC, Kempaiah P, Hittner JB, and Ong'echa JM (2011) Severe malarial anemia: innate immunity and pathogenesis. *Int J Biol Sci*, **7**, 1427-1442.
18. Warimwe GM, Keane TM, Fegan G, Musyoki JN, Newton CR, Pain A, Berriman M, Marsh K, and Bull PC (2009) *Plasmodium falciparum* var gene expression is modified by host immunity. *Proc Natl Acad Sci U S A*, **106**, 21801-21806.

2nd Editorial Decision

05 May 2016

Thank you for the submission of your revised manuscript to EMBO Molecular Medicine. We have now received the enclosed reports from the referees that were asked to re-assess it. As you will see, the reviewers are globally supportive. However, before to move forward, I would like you to address the remaining issue(s) highlighted once more by referee 2. Those are the same initially rose by this referee; while mostly satisfied with the revision, s/he requests additional details (including experimentally). Please proceed with revision as suggested in the report and reply to the referee concerns in a point-by-point letter.

I look forward to seeing a revised form of your manuscript as soon as possible.

***** Reviewer's comments *****

Referee #1 (Remarks):

Improved manuscript, clear data.

Referee #2 (Remarks):

The authors have partially addressed my concern re quantitation by using transformed q-rt-pcr data for the domain analysis. The q-rt-pcr data is now presented as the proportion that each of 6 genes comprises of the sum of the same 6 genes that are predicted by endpoint pcr to be most abundant in a patient. The formula for the transformation seems appropriate except that for the percentage of unannotated transcripts the reader is referred to the dbla tag analysis but nowhere in the text are the unannotated transcripts described except in the legend for EV1 line 917-20. Here it appears they are ectodomains that could not be successfully amplified from the 6 abundant dbla tags, this should be moved to, or repeated in, the methods.

However I also explicitly asked for validation of the endpoint pcr dbla sequence tag counting to be presented, e.g. as scatter graphs plotted against $2^{\text{exp-delta Ct}}$ for the same genes. This is important as the q-rt-pcr analysis depends on the assumption that the endpoint pcr and 454 sequencing correctly identified six abundantly transcribed genes per patient, as these 6 constitute the total var transcripts included in the q-rt-pcr analysis.

The authors have done this analysis but have not provided it as they state on line 485.

"The comparison between patient groups gave almost identical results as those obtained by qPCR adjusted data. However, it should be noted that the two methods of quantification produced slightly different results within a given patient."

If the results are slightly different that is OK but should be shown. If they are very different this could be due to absence in the q-rt-pcr of dbla sequence tags that were present at low levels in the

endpoint pcr, leaving only be a limited, upper range of q-rt-pcr values to plot. Including some q-rt-pcr on low abundance dbla tags would establish the low end of the range and enable readers to assess the validity of the selection of the 6 genes from the endpoint pcr as being abundant.

Alternatively the authors could use their extensive suite of var primers to estimate total levels of exon2 or ups A+B+C in each patient and compare the levels of these indicators of total var transcription to the 6 transcripts predicted to be abundant.

2nd Revision - authors' response

21 May 2016

Referee #2 (Remarks):

Referee: The authors have partially addressed my concern re quantitation by using transformed q-rt-pcr data for the domain analysis. The q-rt-pcr data is now presented as the proportion that each of 6 genes comprises of the sum of the same 6 genes that are predicted by endpoint pcr to be most abundant in a patient. The formula for the transformation seems appropriate except that for the percentage of unannotated transcripts the reader is referred to the dbla tag analysis but nowhere in the text are the unannotated transcripts described except in the legend for EV1 line 917-20. Here it appears they are ectodomains that could not be successfully amplified from the 6 abundant dbla tags, this should be moved to, or repeated in, the methods.

Reply: The description of un-annotated genes is now given in the methods section: "A list of assembled and annotated transcripts is shown in Appendix Figure S1. For some genes it was not possible to amplify the full-length gene or to assemble the full sequence. For these domain un-annotated genes the DBLa-tag sequence information was used to assign the transcript to a var group as follows:"

Referee: However I also explicitly asked for validation of the endpoint pcr dbla sequence tag counting to be presented, e.g. as scatter graphs plotted against $2^{\text{exp-delta Ct}}$ for the same genes. This is important as the q-rt-pcr analysis depends on the assumption that the endpoint pcr and 454 sequencing correctly identified six abundantly transcribed genes per patient, as these 6 constitute the total var transcripts included in the q-rt-pcr analysis. The authors have done this analysis but have not provided it as they state on line 485.

"The comparison between patient groups gave almost identical results as those obtained by qPCR adjusted data. However, it should be noted that the two methods of quantification produced slightly different results within a given patient."

If the results are slightly different that is OK but should be shown. If they are very different this could be due to absence in the q-rt-pcr of dbla sequence tags that were present at low levels in the endpoint pcr, leaving only be a limited, upper range of q-rt-pcr values to plot. Including some q-rt-pcr on low abundance dbla tags would establish the low end of the range and enable readers to assess the validity of the selection of the 6 genes from the endpoint pcr as being abundant.

Alternatively the authors could use their extensive suite of var primers to estimate total levels of exon2 or ups A+B+C in each patient and compare the levels of these indicators of total var transcription to the 6 transcripts predicted to be abundant.

Reply: We agree that the DBLa-tag PCR approach risks under- or overestimating the relative abundance of single var transcript species. Although we have previously designed several primer sets that target different subsets of var genes, these cannot be used to determine the total var transcript abundance with a sensitivity and precision required to check if var species have been missed by the DBLa-tag analysis (degenerate primer to unknown targets will be prone to biases; in previous unpublished work we have failed to design and validate primers targeting exon2 for quantifying "all vars").

We agree that showing a correlation between the qPCR and DBLa-tag quantifications, lends support to the genes chosen for further analysis indeed were the most prominently expressed.

We have therefore added plots (Appendix Figures S2+S3) of qPCR reported transcript abundances vs. DBLa-tag proportions as well as qPCR corrected transcript proportions vs. DBLa-tag proportions. Overall these plots validated subdominant and dominant transcripts. The qPCR correction of the profiles reassigned a median of 21% transcript proportion between the measured genes in each patient (now stated in the Methods section).

The plots also show that the selected genes must be considered as highly expressed based on what has been observed in other in vivo studies of var2csa gene expression in pregnant women (Ndam JID 2005, Ndam PLoS One 2008). Here, var2csa transcript abundances in placental parasites range from dCt 3 to -5 whereas the interquartile range of high binders was 0 to -4, median -1 (dCt values corrected to correspond to the same internal control gene as used for the children - fructose-bisphosphate aldolase vs seryl-tRNA synthetase). The measured transcript abundances of the severe malaria children top6 genes range from dCts 5 to -4.5, and the accumulated top6 transcripts correspond to dCts from -0.3 to -5.5, median -3.1.

Tuikue Ndam NG, Salanti A, Bertin G, Dahlbäck M, Fievet N, Turner L, Gaye A, Theander T, Deloron P. High level of var2csa transcription by Plasmodium falciparum isolated from the placenta. J Infect Dis. 2005 Jul 15;192(2):331-5. Epub 2005 Jun 14. PubMed PMID: 15962229.

Tuikue Ndam N, Bischoff E, Proux C, et al. Plasmodium falciparum Transcriptome Analysis Reveals Pregnancy Malaria Associated Gene Expression. Beeson J, ed. PLoS ONE. 2008;3(3):e1855. doi:10.1371/journal.pone.0001855.

Corresponding Author Name: Thomas Lavstsen

Manuscript Number: EMM-2016-06188